MS-TP-21-21, KA-TP-18-2021, P3H-21-052, IFJPAN-IV-2021-11

# Impact of W and Z Production Data and Compatibility of Neutrino DIS Data in Nuclear Parton Distribution Functions

K.F. Muzakka[1*], P. Duwentäster[1], T.J. Hobbs[2,3,4], T. Ježo[5], M. Klasen[1], K. Kovařík[1], A. Kusina[6], J.G. Morfín[7], F. I. Olness[2], R. Ruiz[6], I. Schienbein[8], J.Y. Yu[8]

**1** Institut für Theoretische Physik, Westfälische Wilhelms-Universität Münster, Wilhelm-Klemm-Straße 9, D-48149 Münster, Germany
**2** Southern Methodist University, Dallas, TX 75275, USA
**3** Jefferson Lab, Newport News, VA 23606, USA
**4** Department of Physics, Illinois Institute of Technology, Chicago, Illinois 60616, USA
**5** Institute for Theoretical Physics, KIT, D-76131 Karlsruhe, Germany
**6** Institute of Nuclear Physics Polish Academy of Sciences, 31342 Kraków, Poland
**7** FNAL, Batavia, IL 60510, USA
**8** Laboratoire de Physique Subatomique et de Cosmologie, Université Grenoble-Alpes, CNRS/IN2P3, 53 avenue des Martyrs, 38026 Grenoble, France

*khoirul.muzakka@uni-muenster.de

July 28, 2021

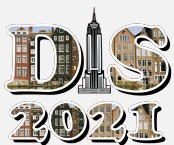

*Proceedings for the XXVIII International Workshop on Deep-Inelastic Scattering and Related Subjects, Stony Brook University, New York, USA, 12-16 April 2021*

## Abstract

**Vector boson production and neutrino deep-inelastic scattering (DIS) data are crucial for constraining the strange quark parton distribution function (PDF) and more generally for flavor decomposition in PDF extractions. We extend the nCTEQ15 nuclear PDFs (nPDFs) by adding the recent $W$ and $Z$ production data from the LHC in a global nPDF fit. The new nPDF set, referred to as nCTEQ15WZ, is used as a starting point for a follow-up study in which we assess the compatibility of neutrino DIS data with charged lepton DIS data. Specifically, we re-analyze neutrino DIS data from NuTeV, Chorus, and CDHSW, as well as dimuon data from CCFR and NuTeV. To scrutinize the level of compatibility, different kinematic regions of the neutrino data are investigated. Fits to the neutrino data alone and a preliminary global fit are performed and compared to nCTEQ15WZ.**

## 1 Introduction

Nuclear parton distribution functions (nPDFs) are key inputs for any factorization-based theory prediction that involve a nucleus in the initial state [1–3]. Due to their non-perturbative nature, nPDFs are determined from a QCD global analysis of various nuclear data, such as charged lepton or neutrino deep-inelastic scattering (DIS), or Drell-Yan (DY) lepton pair production data. Most of these data are sensitive particularly to (anti-)up and (anti-)down quark nPDFs, while the strange quark and gluon nPDFs are traditionally the least constrained and thus have larger uncertainties.

It has been shown in a reweighting study [4] that the recent $W$ and $Z$ boson production data from the LHC [5–11] provide better constraints on the strange and gluon nPDFs. However,

other $W$ boson-mediated processes, such as neutrino DIS and the more exclusive observable of charm dimuon production, are also very sensitive to the strange quark PDF [12, 13].

Including neutrino DIS data in a global fit is not straightforward as these data have been shown to display tensions with the charged lepton DIS data [14–17]. To make the situation even more complicated, while tension was observed in Ref. [14] (which used correlated systematic uncertainties), a separate analysis [18] of $F_2$ and $F_3$ neutrino data from NuTeV, Chorus, and CDHSW (without correlated systematic uncertainties) found compatibility with the charged lepton DIS and DY data. Additionally, another study [19] normalized the integrated differential cross section for each incident neutrino energy $E$ bin to suppress the $E$-dependence fluctuation and, without correlated systematic uncertainties, obtained compatibility. Clearly, there remains important questions to address regarding these data sets.

In this analysis, we review the impact of including $W$ and $Z$ boson production data from LHC on the fitted nPDFs (referred to as nCTEQ15WZ); additional details can be found in [20]. We include neutrino data from NuTeV [21], CDHSW [22], and Chorus [23], as well as charm dimuon data from CCFR and NuTeV [24]. We first analyze the neutrino data separately and then perform a global fit combining all the data sets used in the nCTEQ15WZ analysis with the neutrino data.

## 2 Theoretical Framework and Data Sets

In all our fits, the same nCTEQ15 fitting framework is employed. In particular, the PDFs of a proton in the nuclear medium are parametrized in a polynomial form [1], and the mass number ($A$) dependence is parametrized at the level of the PDF parameters. All theory calculations are performed at next-to-leading order (NLO) in perturbative QCD. The ACOT scheme [25, 26] is used for the treatment of heavy quarks. To minimize higher twist effects and avoid the resonance region, we apply the standard kinematic cuts of $Q > 2$ GeV and $W > 3.5$ GeV for all the DIS data, including the neutrino sets. After these kinematic cuts, the remaining number of data points in the nCTEQ15WZ set is 853, plus 2366 from NuTeV, 929 from CDHSW, 824 from Chorus, and 174 from the charm dimuon data. In the nCTEQ15WZ fit as well as the new fits with neutrino data, 19 PDF parameters are actively varied during the minimization process including: four $u_\nu$ parameters ($a_1^{u_\nu}, a_2^{u_\nu}, a_4^{u_\nu}, a_5^{u_\nu}$), three $d_\nu$ parameters ($a_1^{d_\nu}, a_2^{d_\nu}, a_5^{d_\nu}$), seven gluon parameters ($a_1^g, a_4^g, a_5^g, b_0^g, b_1^g, b_2^g, b_5^g$), two $\bar{u} + \bar{d}$ parameters ($a_1^{\bar{u}+\bar{d}}, a_5^{\bar{u}+\bar{d}},$) and three $s + \bar{s}$ PDF parameters ($a_0^{s+\bar{s}}, a_1^{s+\bar{s}}, a_2^{s+\bar{s}},$). The rest of the PDF parameters are fixed to the same values as those in nCTEQ15 PDFs. We have checked that opening more parameters in the minimization yields little to no improvement in the obtained $\chi^2$ values.

## 3 The nCTEQ15WZ fit

To study the impact of the LHC data on the extracted nPDFs, we performed a global analysis with the nCTEQ15WZ data set. In addition to varying 19 PDF parameters, we also fit three normalization parameters for the LHC $W/Z$ data sets, specifically for i) CMS Run I data, ii) CMS Run 2 data, and iii) ATLAS data.

The result of the fit, in terms of the $\chi^2$ values per number of data points, is reported in Fig.1. The figure shows that an excellent overall fit was obtained, and good descriptions of the data was observed for all experiments [20][1]. The LHC data is well described and no significant tension was observed with the charged lepton DIS and DY data. Considering the fact

---

[1]Note that Fig.1 does not include the $\chi^2$/pt from the three $W/Z$ normalization parameters, as they are all within $\sim 1\sigma$, so that they contribute to the total $\chi^2$ less than three units.

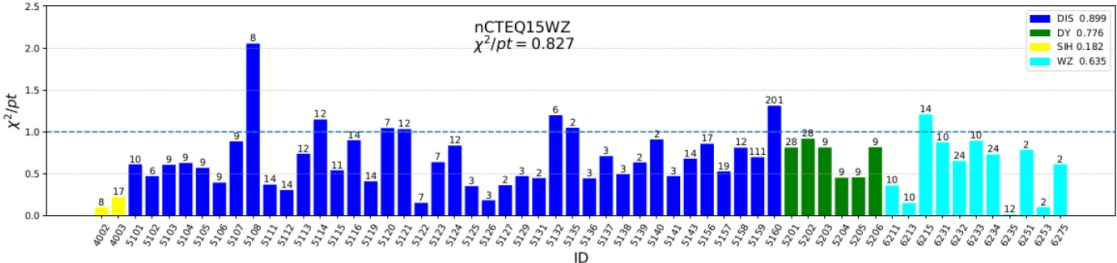

Figure 1: $\chi^2$/pt for the nCTEQ15WZ fit. The data IDs are defined in [20], and the number of data points is indicated on top of each bar. For each process, the corresponding $\chi^2$/pt value is shown in the legend.

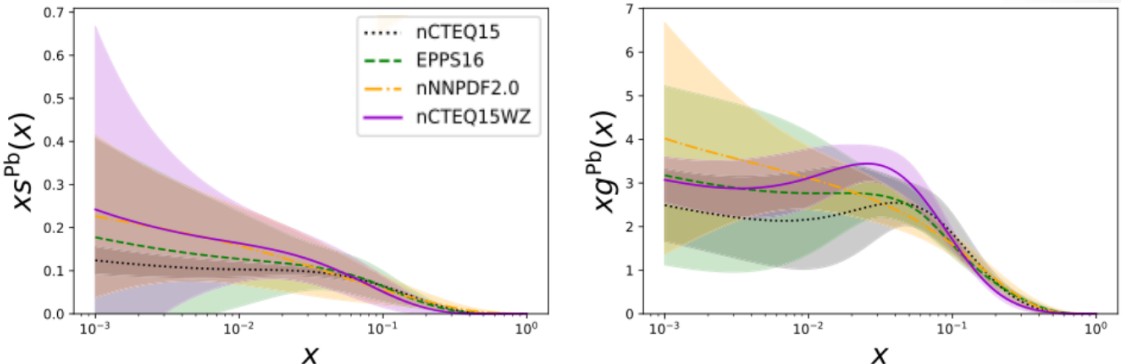

Figure 2: The strange quark and gluon distributions in lead for a selection of nPDFs at $Q = 2$ GeV.

that the LHC $W/Z$ data involves inclusive cross sections of both charged and neutral current processes, this suggests the existence of universal nPDFs that can explain both charged and neutral current processes, at least for this data subset. In light of the compatibility of the above data sets, it will be especially insightful to perform a combined fit of the neutrino data and the data used in the nCTEQ15WZ fit. Recall that the analysis of Ref. [14] found significant tension between these data. We revisit these issues in Sec. 4.

In Fig. 2, we show the strange quark and gluon PDFs from the nCTEQ15WZ fit and selected global fits in the literature: nNNPDF2.0 [3], EPPS16 [2], and nCTEQ15. For the strange quark PDF, nCTEQ15WZ yields a higher distribution at low $x$, suggesting an elevated strange sea ratio, $R_s \equiv (s + \bar{s})/(\bar{u} + \bar{d})$. Interestingly, a similar effect was also observed for the proton PDF using LHC $W/Z$ proton data [27]. Still, the nPDF sets from the different groups agree well within their respective uncertainties. Note that the nCTEQ15WZ $s(x)$ uncertainty band is larger compared to nCTEQ15 because the latter fit constrained the strange quark to be a fixed fraction of $\bar{u} + \bar{d}$ at the input scale. The nCTEQ15WZ fit made no such assumption, and thus it is a more accurate representation of the true $s(x)$ uncertainty. Examining the gluon PDF, we can see a smaller error band compared to nCTEQ15. The improved sensitivity to the gluon PDF can be explained by the high scale of the LHC $W/Z$ data, which probe small $x$ values and are sensitive to higher-order, gluon-initiated corrections.

# 4 Neutrino Fits

In this section, we fit neutrino and dimuon data sets alone (referred to as DimuNeu) and then include these data sets in global analyses together with the nCTEQ15WZ set to study their

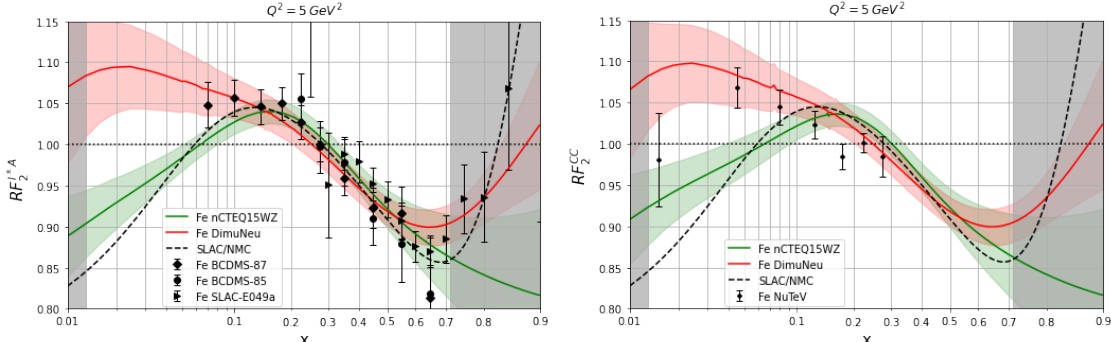

Figure 3: The nuclear ratio $R = F_2^A/F_2^N$ computed with nCTEQ15WZ and DimuNeu for charged lepton (left) and neutrino (right) processes at $Q^2 = 5$ GeV$^2$. $F_2^{CC}$ on the right panel is defined as $F_2^{CC} = (F_2^{\nu A} + F_2^{\bar{\nu}A})/2$. For the charged lepton case we compare with BCDMS [28,29] and SLAC [30] data, and for the neutrino case we compare with NuTeV data [21]. The SLAC/NMC parameterization is taken from [31]. These figures are preliminary and do not fully account for the fitted normalizations.

compatibility and potential tension.

## 4.1 DimuNeu vs. nCTEQ15WZ

To gauge the compatibility between the charged lepton and neutrino data sets, in Fig. 3 we display the predicted nuclear ratio ($R = F_2^A/F_2^N$) using the nCTEQ15WZ and DimuNeu nPDF sets and overlaid with experimental data. The left panel displays the predictions for charged leptons and compared with BCDMS [28,29] and SLAC [30] data and the right panel displays the predictions for neutrinos as compared with NuTeV data [21].

Fig. 3 shows that at a representative case of $Q^2 = 5$ GeV$^2$ both the DimuNeu and nCTEQ15WZ predictions generally agree (within uncertainties) for intermediate $x$, but diverge from each other for $x \lesssim 0.1$, and to a lesser extent in the high $x$ region, i.e., for $x \gtrsim 0.6$. Specifically, the influence of the neutrino data on the DimuNeu fit with $Q^2 > 4$ GeV$^2$ is to extend the anti-shadowing region to smaller $x$ values, and the usual shadowing behavior is not evident.

## 4.2 Compatibility Test

We now assess the compatibility of the nCTEQ15WZ data set ($S$) with the individual neutrino data sets ($\bar{S}$) using the hypothesis-testing method. Data sets $S$ and $\bar{S}$ are compatible if the following two conditions are satisfied: i) When the new data set ($\bar{S}$) is included, the $\chi^2$ increase of the original nCTEQ15WZ data set ($\Delta\chi^2_S$) is less than some tolerance $T$. ii) The combined fit can describe all the data (both $S$ and $\bar{S}$) within the 90% confidence level. For the first condition, we choose a tolerance of $T=45$. The second condition can be checked by examining the percentile $p_{\bar{S}}$.

Table 1 shows the result of individually including each new data set ($\bar{S}$) into the fit with the nCTEQ15WZ data ($S$). We observe that all the neutrino data sets lead to a noticeable $\chi^2$ increase for the nCTEQ15WZ data ($\Delta\chi^2_S$), and only the Chorus and CDHSW data marginally satisfy our compatibility criteria.

Examining these results in more detail, we find that the bulk of the tensions arise from the small $x$ region, where we observed the divergence of the nCTEQ15WZ and DimuNeu results in Fig. 3. To examine the impact of the small $x$ region, we repeat our analysis with a cut of

| $\bar{S}$ | $\Delta\chi^2_S$ | N | $p_{\bar{S}}(\%)$ |
|---|---|---|---|
| Chorus | 17 | 824 | 83.8 |
| CDHSW | 44 | 929 | 0.0 |
| NuTeV | **92** | 2136 | **93.5** |
| DimuNeu | **106** | 4063 | **99.2** |

Table 1: $\Delta\chi^2_S$ and the percentiles $p_{\bar{S}}(\%)$ after including the $\bar{S}$ neutrino data sets.

| $\bar{S}$ | $\Delta\chi^2_S$ | N | $p_{\bar{S}}(\%)$ |
|---|---|---|---|
| Chorus$_{[x>0.1]}$ | 9 | 738 | 81.7 |
| CDHSW$_{[x>0.1]}$ | 26 | 845 | 0.0 |
| NuTeV$_{[x>0.1]}$ | 17 | 1737 | 74.6 |
| DimuNeu$_{[x>0.1]}$ | 28 | 3494 | 66.0 |

Table 2: Same as Table 1, but with the $x \geq 0.1$ cut imposed.

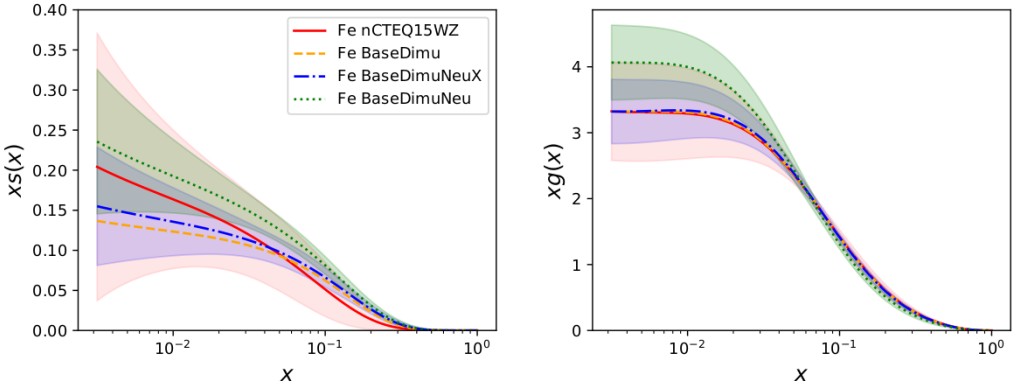

Figure 4: We show the strange quark and gluon PDFs from the combined fits at $Q = 2$ GeV.

$x \geq 0.1$ on the neutrino data. The results of this study are shown in Table 2. The difference is striking and we see that now all the neutrino data sets ($\bar{S}$) satisfy our compatibility criteria.

While we have succeeded in identifying the kinematic region which generates the tension in our fits, additional work is required to find the underlying cause. There are a number of potential sources including: a difference in shadowing of charged lepton and neutrino nucleus scattering, non-perturbative contributions from nuclear effects, higher-twist terms (the small $x$ data typically lies at lower $Q$ values), and the PDF fitting methodology (including assumptions about the PDF parameterization, isospin symmetry, the strange and gluon distributions). Additionally, there could be a combination of multiple factors. Further investigations are ongoing, but we perform a preliminary combined fit in the following section.

## 4.3 The Combined Fit

Having identified the region that generates the tension between data sets, we perform a global nPDF fit including all the data sets (nCTEQ15WZ, dimuon, and neutrino, referred to as BaseDimuNeu). To see the impact of low $x$ neutrino data, we also perform another fit with all the above data, but impose a $x \geq 0.1$ cut on the neutrino data (referred to as BaseDimuNeuX).

The results of these fits are displayed in Fig. 4 for the strange quark and gluon PDFs, as these are most impacted by the new data. For the strange quark PDF, we see that both the nCTEQ15WZ and BaseDimuNeu fits yield comparably large distributions in the small $x$ region. In contrast, both the BaseDimu and BaseDimuNeuX fits prefer a smaller strange PDF at low $x$. The comparison of BaseDimuNeu and BaseDimuNeuX clearly highlights the impact of the low $x$ ($x \leq 0.1$) neutrino data on the strange quark.

Turning to the gluon PDF (Fig. 4, right) we observe that all the fits approximately coincide except for BaseDimuNeu which, like the strange quark, yields an increased PDF in the small $x$ region. Again, this points to the strong influence of the low $x$ ($x \leq 0.1$) neutrino data. Fig. 4 also displays the uncertainty bands that show both the increase of the uncertainty towards

lower $x$ values and the general reduction of the uncertainty as we add new data to the fits.

## 5 Conclusion

We have presented the results from our recent PDF analysis (nCTEQ15WZ), which includes $W$ and $Z$ boson production data from the LHC, and have extended this fit to include new data from neutrino DIS experiments. Specifically, our extended analysis includes inclusive cross section data from NuTeV, CDHSW, and Chorus, as well as charm dimuon data from CCFR and NuTeV.

Both LHC $W/Z$ data and the neutrino data impose strong constraints on the strange quark and gluon PDFs. We are able to obtain good fits to the data sets as measured by their $\chi^2$ values. However, there are some tensions that are most evident by comparing fits (BaseDimuNeu and BaseDimuNeuX) with and without the low $x$ ($x \leq 0.1$) neutrino data. This difference is mostly reflected in the resulting strange PDF at low $x$ values.

We have conducted this study examining the $\chi^2$ for individual data and individual processes (Fig. 1), together with the nuclear ratios (Fig. 3) and the compatibility criteria (Sec. 4.2). This powerful combination of tools allows us to incisively analyze these data sets and identify the source and impact of the tensions. This analysis represents an important step forward as improved nPDFs yield both precision predictions for hadronic process and an enhanced understanding of the underlying nuclear interactions.

## Acknowledgements

We would like to thank Alberto Accardi, Nasim Derakhshanian, Cynthia Keppel, Chloé Leger, Florian Lyonnet, and Peter Risse for useful discussions. The work of M.K., K.K. and K.F.M. was funded by the Deutsche Forschungsgemeinschaft (DFG, German Research Foundation) through the Research Training Group GRK 2149. P.D., M.K. and K.K. also acknowledge the support of the DFG through project-id 273811115 – SFB 1225. A.K. and R.R. acknowledge the support of Narodowe Centrum Nauki under Grant No 2019/34/E/ST2/00186. The work of T.J. was supported by the Deutsche Forschungsgemeinschaft (DFG, German Research Foundation) under grant 396021762 - TRR 257. F.O. acknowledges support through US DOE grant No. DE-SC0010129 and the National Science Foundation under Grant No. NSF PHY-1748958. The work of I.S. was supported by the French CNRS via the IN2P3 project GLUE@NLO.

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
