# Peer review of "Impact of W and Z Production Data and Compatibility of Neutrino DIS Data in Nuclear Parton Density Extraction"

_SciPost Physics Proceedings_

## Round 1 · Referee Report · Anonymous (Referee 1) · 2022-2-28

Report

The paper, "Impact of W and Z Production Data and Compatibility of Neutrino
DIS Data in Nuclear Parton Distribution Functions", presents the results from a PDF analysis (nCTEQ15WZ), which includes W and Z boson production data from the LHC, and have extended this fit to include new data from neutrino DIS experiments, as well as charm dimuon data. The discrepancy between the fits presented here in the high and low x regions are interesting and are evident in the strange PDFs at low x values.

There are two comments that should be addressed. In section 4.2 I see that you have written "...lower Q values." and in the caption to Fig.4 you write "Q = 2 GeV". Throughout the rest of the paper you use Q^2 and GeV^s, is that what was intended to be written in section 4? Also, what is the reason for choosing a tolerance of 45 in section 4.2?

---

## Editorial Decision

resubmitted